# An Evaluation of a Pesticide Training Program to Reduce Pesticide Exposure and Enhance Safety among Female Farmworkers in Nan, Thailand

**DOI:** 10.3390/ijerph20176635

**Published:** 2023-08-24

**Authors:** Thanawat Rattanawitoon, Wattasit Siriwong, Derek Shendell, Nancy Fiedler, Mark Gregory Robson

**Affiliations:** 1Department of Environmental and Occupational Health and Justice, Rutgers School of Public Health, Piscataway, NJ 08854, USA; shendedg@sph.rutgers.edu (D.S.); nfiedler@eohsi.rutgers.edu (N.F.); 2College of Public Health Sciences, Chulalongkorn University, Institute Building 3 (10th–11th Floor), Chulalongkorn soi 62, Phyathai Rd., Bangkok 10330, Thailand; wattasit.s@chula.ac.th; 3NJ Safe Schools Program, Rutgers School of Public Health, Piscataway, NJ 08854, USA; 4Rutgers Environmental and Occupational Health Sciences Institute, Piscataway, NJ 08854, USA; mark.robson@rutgers.edu; 5Department of Plant Biology, Rutgers University, New Brunswick, NJ 08901, USA

**Keywords:** pesticide exposure, female farmworkers, pesticide training program

## Abstract

Background: Although exposure to chemical pesticides is known to cause negative effects on human health, farmers in Ban Luang, Nan, Thailand, continue to use them regularly to protect crops. This study focused on mothers who were engaged in farm tasks and had children between the ages of 0 to 72 months, with the objective of reducing pesticide exposure. Methods: This study was conducted from May 2020 to October 2020 in the Ban Fa and Ban Phi sub-districts in Ban Luang due to the high use of pesticides in these areas. A systematic random sampling technique was used to recruit 78 mothers exposed to pesticides. Thirty-nine mothers from Ban Fa district were randomly assigned to the intervention group and 39 from Ban Phi to the control group over a 3-month period. This study applied a pesticide behavioral change training program for the intervention group. To assess the effectiveness of the program, the study compared the results of a questionnaire regarding knowledge, attitude, and practice (KAP) and health beliefs related to pesticide exposure as well as the levels of acetylcholinesterase (AChE) and butyryl cholinesterase (BChE) enzymes, biomarkers of exposure to pesticides, before and after the intervention using ANCOVA statistical test. Furthermore, to evaluate the effectiveness of the intervention program, a paired *t*-test was used to investigate the in-home pesticide safety assessment. Results: After the intervention, we observed no significant change in AChE; however, a significant improvement in BChE (*p* < 0.05), a marker of short-term recovery, was observed. Pesticides can cause a reduction in AChE and BChE, however, after eliminating pesticides, BChE takes a shorter time (about 30–50 days) to recover than AChE (around 90–120 days). Therefore, increases in the measured concentrations of AChE and/or BChE suggest the presence of less chemicals from pesticides in the human body. The study also found a significant improvement in KAP and beliefs about chemical pesticide exposure after the intervention (*p* < 0.05). Furthermore, using a paired *t*-test, we found a significant increase in pesticide safety practices (*p* < 0.05) in the intervention group and a borderline significant increase regarding in-home safety (*p* = 0.051) in the control group. Conclusions: Based on the results, the constructs of the intervention program were effective and could be applied in other agricultural areas in less developed countries. However, due to time limitations during the COVID-19 pandemic, further studies should be conducted to enable data collection over a longer time, with a larger number of subjects providing the ChE levels for the non-agricultural season.

## 1. Background

Pesticides are widely used in agriculture to protect crops from pests and to increase yields [1]. However, pesticides also cause negative effects on human health [2,3,4,5,6,7] and the environment [8,9]. Although the adverse outcomes from exposure to pesticides have been documented, the use of pesticides in agriculture has increased worldwide, including in Thailand. Thailand is an agricultural country and reported registered farm households account for 34.4% of total households [10]. This country ranked fourth on the pesticide consumption among 15 Asian countries [11], and had reported a four-fold increase in pesticide consumption between 1990 and 2012. Despite the Thai government’s implementation of domestic pesticide regulations, the use of pesticides has continued to grow and it has been determined as one of the major public health issues in the country [12].

The heavy use of pesticides has been reported in many provinces in Thailand, including Ban Luang district, Nan [13]. Ban Luang is one of the districts in Nan that relies heavily on agricultural products [14]. According to agricultural data, the greatest proportion of land use was in agriculture, with about 32.0% or 12,125 ha and villagers grew mostly maize at around 53.0%, followed by rice and tamarind with 13.9% and 11.0%, respectively [15]. This district ranked as the second highest proportion of farm households registered (92.0%) in Nan in 2018 [16]. The amounts of the pesticides used have not been officially reported; however, farmworkers have reported using pesticides and fertilizers on their crops [17,18]. Furthermore, an agricultural extension specialist has reported on how female farmworkers play an equally important role on farms as men, e.g., by helping families to cultivate their land [17]. Women are more vulnerable to pesticides than men due to a greater sensitivity of some physical characteristics, such as a higher percentage of body fat enabling them to accumulate pesticides, their hormonally sensitive tissues, and their productive age [19,20,21]. Women also have a greater chance to have contact with children. Consequently, exposure to pesticides leads them to negative health outcomes including skin and eye irritation, dizziness, headache, a shortened length of gestation and breast cancer [22,23,24]. In addition, they are more likely to distribute pesticide residues at home to other family members [6], particularly children, resulting in neurological problems [5,23].

The use of pesticides has led to public health issues in Ban Luang. Ban Luang hospital revealed patients were sick from exposure to pesticides, reporting 22.6 per 100,000 population [25]. At the same year, Ban Luang District Public Health Office reported about 73.0% of farmers (n = 703) who were tested for serum cholinesterase (SChE) using reactive papers, had risky and unsafe levels of exposure to pesticide [26]. Moreover, inappropriate behaviors, such as not washing work clothes after spraying pesticide, using the same work clothes for more than one day, not separating work clothes before washing, eating food in or near a farm and not always using PPE, were observed by an agricultural extension specialist [17].

Despite pesticide exposure being recognized as a significant public health issue in Ban Lung, there is a lack of studies to minimize exposure, particularly concerning behavioral change training programs for women. Consequently, there is a need to create a pesticide safety program to reduce the exposures and to enhance the appropriate behaviors of women due to their active roles in agriculture and home activities [17,27]. The main purpose of this study is to create an effective pesticide safety training program applying the theories of knowledge, attitude, and practice (KAP) and the health belief model (HBM) collaborating with stakeholders in these areas.

## 2. Objectives

The objectives of this study are to develop an integrated community-based training program and to assess its effectiveness in increasing the levels of AChE and BChE and in suggesting things such as lower exposures to chemical pesticides, the KAP, health beliefs, and appropriate pesticide behaviors for farm mothers.

## 3. Methods

Study design and study areas: This study was conducted from May 2020 to October 2020 in Ban Luang, Nan, Thailand. It consisted of two phases, including a preparatory and implementation phase. Ban Fa sub-district, with 8 villages, was purposefully selected as the intervention area due to the high amount of pesticide use and pesticide-related health problems found [17,18,26]. Whereas Ban Phi sub-district, with five villages, was purposefully selected as a control area because this area used similar levels of pesticides and had health complaints. Both sub-districts are comparable due to the similarity of a year-round growing season patterns of crops such as maize, rice, tamarind, longan and mango [28], as well as demographic characteristics. The distance between Ban Fa and Ban Phi sub-districts is approximately eleven kilometers [29].

Study population: The study focused on mothers with children aged 0–72 months as these people tend to work on their farms [27,30,31,32] and can distribute pesticide residues to their family members [17,27], especially children, resulting in adverse health outcomes [5,22,23]. To begin the study, we gathered data regarding mothers living in the study areas using a report which was provided by the Nan Public Health Provincial Office [25]. Two hundred twenty-six potential mothers, including 90 and 136 mothers from Ban Fa and Ban Phi sub-districts, respectively, were recruited. Afterward, the research team distributed flyers in the villages. Eventually, 175 out of the 226 participants responded to the researcher and they were appointed for the first recruitment. Women could participate in the study if they were healthy, of the age between 18–44 years, had children aged 0–72 months, lived in Ban Fa and Ban Phi from June 2019 to October 2020, and engaged in farm tasks. They also needed to be literate and be able to communicate with people, as well as be willing to participate in the study. Women who had pre-existing health conditions, such as diabetes, alcoholism, drug addiction, renal failure, liver disease, and cancer, as well as pregnant women, were excluded. Consequently, 139 mothers who met the inclusion criteria, including 78 from Ban Fa and 61 from Ban Phi, were included.

Sample size calculation: Calculations based on comparisons of the two population means for continuous outcomes were employed [33]. The results of the quantitative variables, such as KAP scores and ChE levels, were approximately normally distributed; [34]. This study used protective behavior scores from the previous study, which indicated an increase and difference in the mean scores of protective behaviors between intervention and control groups of 3.73 points [35]. Consequently, with a 10% dropout rate, the sample sizes were 39 for each intervention and control group, with a total of 78 participants. To obtain the sample sizes for each group, a systematic random sampling was utilized with eight villages in Ban Fa and five villages in Ban Phi. In addition, we randomly selected the participants for the two groups using named lists of 139 mothers. Of these, 39 participants of each group were blinded and completed the study.

### 3.1. Procedure and Study Plan

The researcher categorized the study plan into a preparatory and implementation phase. The details for each stage are presented as follows.

#### 3.1.1. Program Development

This study was conducted from May 2020 to July 2020. Due to the pandemic, the researchers could only go to the study areas in June 2020; otherwise, researchers had virtual meetings with the team in the community. Before conducting the study, the researcher trained ten village health volunteers (VHVs) from the study areas on how to use the questionnaire regarding KAP and health beliefs of pesticide exposure and in-home pesticide safety assessment. Afterward, the team collected the questionnaires and the assessments during the last week of July 2020 and analyzed the questions to create the intervention program.

#### 3.1.2. Implementation Programs

The 3-month intervention program was created based on the results of the questionnaire. The intervention and control programs were implemented from August 2020 to October 2020. The activities for the participants in each group are shown in Figure 1. The contents of a 1-day pesticide training program included general knowledge about pesticides, routes and exposure pathways, pesticide residues, overall hazards of pesticides, the “dos” and “don’ts” while working on a farm with pesticides, routine decontamination after exposure to pesticides, and the proper use of PPE. As for the follow-up visits, the study employed mobile phones using the LINE application or trained VHVs. Line is an instant messenger application on mobile phones or computers which allow users to send messages, short VDOs, pictures, and perform audio or video calls [36]. This assisted the researchers in distributing pesticide-related information. Due to COVID-19, the majority of the participants agreed to use the Line application to receive and respond to the materials individually; however, a few of them did not have a smartphone, in which case the study used trained VHVs. The topics during the visits consisted of general knowledge about pesticides, self-risk assessment of pesticide exposure, how people become exposed to pesticides, pesticide safety and decontamination after exposure to pesticides. To discuss the issues found and encourage villagers to use the PPE, the research team and participants attended the community meeting once a month. In addition, the study distributed knowledge about pesticide safety exposure through local broadcasting media once a month. With the intervention program, the study also collaborated with the stakeholders in the community, including villagers, village leaders (VLS), VHVs, agricultural extension officers, public health workers and with community resources including Ban Fa subdistrict-health promoting hospital (SD-HPH), Ban Luang hospital, Ban Luang District Health Office, temples, schools, and local administrative organizations. However, Ban Phi did not receive any of these activities.

To test the efficacy of the pesticide intervention group, the study applied the attention control program (Figure 2), including a one-day training program about in-home safety, and provided a manual titled “clean home, good health, and happy life” provided by the Thai Ministry of Public Health (MoPH) [37]. The contents of the in-home safety training included information regarding in-home environmental hazards, in-home safety practices, proper waste management, indoor air and proper ventilation, and injuries and illnesses related to in-home activities. This program was used mainly to assess the effectiveness of the intervention program [38,39] by comparing the results of the attention and intervention programs in the control group. The significant improvement of the participants who participated in the attention program in the control group, and lack of significant difference from the intervention program, will assure the effectiveness of the intervention program. In addition, this helped the participants to engage in the study.

An assessment of the intervention program: the study compared the results of the questionnaire, the enzymes AChE and BChE levels, and in-home pesticide safety assessments before and after the intervention between groups. AChE and BChE indicate pesticides including organophosphates (OPs) and carbamates (CBs) in humans [40]. AChE is found in motor neurons and sensory neurons, presenting in red blood cells and the nervous system. BChE is found mostly in plasma and is produced by the liver [41,42]. Exposure to OPs can inhibit the two enzymes and result in low values, however, when people are not exposed, the AChE and BChE will represent normal values in the body [43,44].

Data collection and specimen collection: Seventy-eight mothers were interviewed using a questionnaire that included KAP and health beliefs of pesticide exposure. Blood samples were drawn two times in the last weeks of July 2020 and October 2020 due to the durations of the use of OPs and CBs in crops. In terms of the questionnaire, the KAP and health belief questions came from the study titled SAWASDEE birth cohort Thailand [45] and was used in conjunction with a cohort study in Chiang Mai, in the north of Thailand [46].

Part one included demographics such as age, marital status, educational attainment, average income, agricultural equipment storage, fertilizers and pesticides, and distance from the field to home. It also included the occupational characteristics of the participants, particularly farm activities.

Part two was applied to examine the knowledge, attitudes and practices that influence pesticide exposure among mothers working in high pesticide use areas. Knowledge related to pesticide exposure comprised 25 questions and each question provided three options, including yes, no, and uncertain. Attitudes concerning pesticide issues and human health consisted of 16 questions, including 7 positive and 9 negative questions. The answers were scored agree, disagree, and uncertain. The section on practices consisted of 18 questions including 10 positive and 8 negative questions related to pesticide safety practices. The answers were scored as thus: always, usually, sometimes, and never.

Part three related to health beliefs from the HBM and was used to assess material perceptions. There were six questions on each of the following: perceived susceptibility, perceived severity, perceived benefits, and perceived barriers from the previous studies [47,48,49,50,51].

The content validity of the questions was verified. For the interview, the trained VHVs visited the participants at their homes. The participants then spent about 20 min to complete the consent form. Afterward, the team spent about 30 min interviewing the participants using the questionnaire.

In addition, this study applied the in-home pesticide safety assessment. This tool helped the researcher obtain insight on the factors causing pesticide exposure in participants’ homes. The questions were modified from the U.S. Agricultural Health Study (AHS) and other previous studies [47,48,52] and the content validity of the questions was verified. This tool comprised two parts, including 12 questions of in-home pesticide safety and 12 questions of safety in the home to assess the attention control program. Home data collection was conducted two times using trained VHVs during the last weeks of July 2020 and October 2020. Each visit took about 30 min.

Regarding the blood collection, the study applied the Test-mate ChE Cholinesterase Test System (Model 400) with the Ellman method [44,53] to determine organophosphates (OPs) and carbamates (CBs) exposure using AChE and BChE levels in blood from a cleaned finger. Before the blood test, the researcher also used the questionnaire to investigate whether the participants were exposed to pesticides in the past few months. The participants were then appointed at Ban Fa and Ban Phi SD-HPHs for blood collection using a nurse practitioner, taking about 12 min to complete the process for each participant. The results of ChE levels were presented as a continuous variable, units per milliliter (U/mL) [44]. This was conducted two times in the last weeks of July 2020 and October 2020.

After collection, the data were stored in Excel files and SPSS version 27 with computer password protection, so only the researcher could access it.

### 3.2. Data Analysis

We applied descriptive statistics to calculate demographic and occupational characteristics and the safety in-home assessment as frequency, percentage, mean, and standard deviation. This study also used a Chi-square test, Fisher’s exact test and an independent *t*-test to examine the demographic and occupational factors between groups before the intervention. Furthermore, the Kolmogorov–Smirnov (KS) test was used to test distribution of quantitative variables, i.e., outcomes, including AChE and BChE levels and KAP and health beliefs scores, as well as those of this study found these variables to be normally distributed. An analysis of covariance (ANCOVA) test was used to determine the differences in the levels of AChE and BChE, the scores of KAP and health beliefs derived from HBM between groups, before and after the intervention. Moreover, to test the efficacy of the intervention program, the study used a paired *t*-test to examine the difference in the mean scores of the in-home pesticide safety assessment within groups after intervention.

## 4. Results

### 4.1. Comparison of the Variables before the Intervention

#### 4.1.1. The Participants Characteristics

In total, 78 female farmworkers participated in the study with 39 participants in each of the intervention and control groups. The average age was 36.47 ± 5.58 years and the participants in the intervention group were slightly younger than the control group (35.7 ± 4.7 and 37.2 ± 6.3 years, respectively). Most of the participants were married. More than half (53.8%) of the participants in the intervention group had completed education through high school. Most of the participants in the two areas had a monthly income lower than about 6000 baht or $202 USD [54]. Most of the participants, 66.7% for the intervention and 79.5% for the control group, did not have a designated safe room to store agricultural equipment and pesticides. Approximately four out of five participants in both areas lived within 50 m of the fields, or gardens.

In terms of the occupation characteristics, participants in both groups had engaged in at least one farm task over the most recent months and exhibited similar patterns while engaging in those farm tasks. For example, the participants in the intervention (76.9%) and the control groups (64.1%) had sprayed or helped carry the rubber sprayer hose. More than half of the participants (56.4%) in the intervention group had handled pesticides, which was higher than the participants in the control group (41.0%). Overall, nearly one-in-three participants mixed the pesticides (32.1%).

From this study, there were no statistically significant differences in demographic and occupational factors between the two groups before the intervention (see Table 1). This implies the two study areas were similar in terms of overall demographic and occupational characteristics.

#### 4.1.2. ChE Activity Levels

Before the intervention, this study found that the mean of AChE in the intervention group was equal to the mean of AChE in the control group (2.7 ± 0.3 U/mL). The mean of BChE in the control group (1.7 ± 0.3 U/mL) was similar to the mean of BChE in the intervention group (1.6 ± 0.3 U/mL). There was no statistically significant difference in terms of AChE or BChE between the two groups before the intervention (Table 2).

#### 4.1.3. KAP and Health Belief Scores between Two Groups

Table 3 reports the KAP and health belief scores between the two groups before the intervention. Overall, the participants in the intervention group had higher scores in the knowledge, attitude, and practice categories compared with the control group. However, there was no statistically significant difference of KAP between the two groups before the intervention. In terms of health beliefs scores, this study suggests that the participants in the intervention group had slightly higher but not significantly different perceived susceptibility, perceived benefits, and perceived barriers scores than those in the control group. Participants in the control group had slightly higher but not statistically different perceived severity scores compared with the intervention group. This study found there was no statistically significant difference in perception between the groups before the intervention. This indicates that KAP and perceptions were not different between the two groups.

#### 4.1.4. The Results of the Questionnaires for Program Development

After conducting the questionnaire before the intervention, the study revealed that the participants showed a lack of comprehensive knowledge regarding adverse health issues such as obesity, slower learning, heart attack/stroke and abdominal pain/diarrhea as the potential impacts of pesticide exposure. The participants also had a negative attitude towards pesticide exposure, especially the linkage between the amount of pesticide and adverse health outcomes and the issues of not understanding how vulnerable populations are exposed to pesticides. Some of them also believed that adults are more vulnerable to pesticides than children and babies. As for the pesticide safety practice category, the participants did not practice appropriate activities while working on their farms, e.g., drinking beverages and eating food, using limes to wash their hands before eating, wearing the same work clothes for more than one day, wearing their unwashed clothes in their homes, and washing their work clothes with other family members’ clothes.

With respect to the health beliefs from the HBM category due to perceived susceptibility, some participants believed that exposure to pesticides can only cause health issues for the farmers. Regarding the perceived severity category, some participants did not believe short-term exposure to pesticide could result in health consequences and more than half believed that repeated exposure to pesticides for an extended period would build tolerance against the negative health consequences. As for the perceived benefits category, some of the participants did not believe that taking a bath after working on their farm would be helpful to mitigate pesticide exposure. Regarding the perceived barriers category, some of the participants did not wear full PPE due to the discomfort they experienced.

### 4.2. Comparison of the Variables after the Intervention

#### 4.2.1. ChE Activity Levels

After the intervention (see Table 4), the study showed that the mean of AChE slightly increased from 2.7 ± 0.3 U/mL to 2.8± 0.3 U/mL in the intervention group and the mean of AChE was identical (~2.7 ± 0.3 U/mL for both before and after the intervention) in the control group. In addition, after adjusting the levels of AChE before the intervention, there was no statistically significant difference in the levels of AChE between groups after the intervention (*p* > 0.3). This indicates that levels of AChE activity among the participants were not affected while conducting the study.

In terms of BChE levels, the study found an increase in BChE between 1.6 ± 0.3 U/mL and 1.8 ± 0.2 U/mL in the intervention group. However, the mean of BChE was almost identical (~1.7 ± 0.3 U/mL both before and after the intervention) in the control group. Additionally, after adjusting the levels of BChE before the intervention, the study found that there was a statistically significant difference in the levels of BChE between groups after the intervention (*p* < 0.00).

#### 4.2.2. KAP and Health Belief Scores between Two Groups

Table 5 presents the intervention effects and the effectiveness of the program on the KAP and health beliefs derived from HBM between each group after the intervention. After adjusting the scores of the pre-intervention, there was a statistically significant difference in the scores on knowledge between the groups after the intervention (*p* < 0.00). About 10.0% of all variances in knowledge scores were attributable to the pesticide safety training program.

When it comes to attitudes, there was a statistically significant difference in the scores of attitudes between the groups after the intervention (*p* < 0.00), after adjusting the scores of the pre-intervention. Approximately 31.0% of all variances in attitude scores were attributable to the pesticide safety training program.

With respect to practices, after adjusting the scores of practices before the intervention, there was a statistically significant difference in the scores of practices between the groups after the intervention (*p* < 0.00). About 27.0% of all variances in practice scores are attributable to the pesticide safety training program.

Regarding perceived susceptibility, there was a statistically significant difference in the scores of perceived susceptibilities between the groups after the intervention (*p* < 0.00), after adjusting the scores of the pre-intervention. Approximately 33.0% of all variances in perceived susceptibility scores were attributable to the pesticide safety training program.

This study also found that there was a statistically significant difference in the scores of perceived severity between the groups after the intervention (*p* < 0.00), after adjusting the scores of perceived severity before the intervention. About 38.0% of all variances in perceived severity scores were attributable to the pesticide safety training program.

According to perceived benefits, there was a statistically significant difference in the scores of perceived benefits between the groups after the intervention (*p* < 0.00), after adjusting the scores of the pre-intervention. Approximately 34.0% of all variances in perceived benefits were attributable to the pesticide safety training program.

With respect to perceived barriers, there was a statistically significant difference in the scores of perceived barriers between the groups after the intervention (*p* < 0.00), after adjusting the scores before the intervention. Around 26.0% of all variances in perceived barriers were attributable to the pesticide safety training program.

### 4.3. The Evaluation of Participant Behaviors at Home

The study applied pesticide safety checklists to evaluate the conditions of the participants homes in the intervention group. The in-home safety checklists for the control group were used in the attention control program. Table 6 shows that there was a statistically significant increase (*p* < 0.05) in the pesticide safety scores (from 8.0 ± 1.3 to 8.5 ± 1.0) in the intervention group. However, there was no statistically significant difference of mean scores of the safety in-home. In terms of the in-home safety assessment from the control group, the study found that the mean score of in-home safety increased from 7.3 ± 1.1 to 7.7 ± 1.1 with a borderline statistical significance level at *p* = 0.051. There was also no significant difference in pesticide safety scores in this group.

## 5. Discussion

The study revealed that a pesticide training program collaborating with community resources appears to be effective and can increase knowledge, attitudes, practices, and perceptions derived from HBM as well as the levels of BChE among mothers in the intervention groups. In terms of ChE activity, after adjusting the ChE activity, there were no statistically significant differences in the levels of AChE between the groups after the intervention. However, the study found that there was a statistically significant difference in the levels of BChE between the groups after the intervention. The resulting improvement of BChE was similar to the results of a study conducted by Nganchamuang [55] in northeastern Thailand. That study applied an agrochemical safety program and collected data three times at the baseline, and at 5 and 8 months after intervention. After receiving the program for 5 months, the mean of AChE was significantly improved from 2.7 ± 0.9 U/mL to 3.3 ± 0.6 U/mL. In addition, the study reported an improvement in the mean of BChE between 1.5 ± 0.4 U/mL and 1.6 ± 0.4 U/mL. This study, however, was conducted over a 3-month period, in fact, the activity of AChE can recover in about 90–120 days and BChE may recover around 50 days [56], therefore, it is possible that the time duration may impact the levels of AChE. The study by Nganchamuang [55] took five months to observe statistically significant increases in AChE and BChE levels. Another one-year study by Suwarng [57] used whole blood to determine the levels of ChE after the intervention program. This study found that the ChE level of the farmer group increased from 2.37 ± 0.57 U/mL (pre-program) to 2.98 ± 0.5 U/mL (post-program). The pesticide intervention programs could help to increase the levels of ChE; however, the magnitudes of the changes were different. We feel that a longer period for the intervention program would likely increase a change of the levels of ChE. Further studies would likely increase the time for data collection and collect a baseline of participants before exposure to pesticides.

From the study, the appropriate pesticide safety practices were observed and reported after the intervention. For example, the participants reported using the full PPE and some of them brought filled water bottles to the farms for use with soap to wash their hands instead of using limes. Some of them would only drink and eat during the breaks and began storing pesticides and farm equipment in safer areas. They also provided a basket to separate work clothes outside of their homes, and they washed their clothes after work every day. Furthermore, they washed their work shoes and left them on the bamboo sticks outside their homes. Some began washing vegetables and fruits using salt or baking soda and rinsing thoroughly with water. Therefore, the changes in behavior resulting from the intervention program could affect the levels of AChE and BChE.

After the intervention, there was a statistically significant difference in the scores of KAP between the groups after the intervention (*p* < 0.00), after adjusting the scores of the pre-intervention. In terms of the statistically significant improvement of KAP, the results of this study are consistent with other studies. Quandt et al. [58] conducted a study in the U.S. using lay promotors for health education and reported an improvement of knowledge among migrant farmers. Additionally, studies in Thailand found significant improvements of knowledge after receiving the intervention pesticide safety training program [35,47,55,59]. With respect to attitude, other studies have reported comparable results. Gesesew et al. [60] reported an increase in knowledge, attitudes, and safety behaviors while using pesticides after implementing a health education program. A previous study in the U.S. also found a positive attitude towards wearing gloves while working on strawberry farms and taking clothes off before entering a home among its participants after the relevant intervention [61]. In addition, studies in Thailand revealed a statistically significant increase in attitudes about pesticide safety among farmers after receiving a pesticide training program [47,55,59]. Regarding pesticide safety practices, this study found a statistically significant improvement in practices. The results are also consistent with other studies. Studies in the U.S. have found an improvement in the practices of pesticide use among farmers after receiving a health education program [58,61,62]. The previous studies in Thailand by Nganchamuang [55], Phataraphom [35], Raksanam [47] and Wongwichit [59] found an improvement of pesticide safety practices among farmers after receiving the intervention.

The studies revealed that implementing strategies or health educational programs enhances knowledge and attitude and that improved knowledge and improved attitude may enhance self-practice. Consequently, these actions may lead to enhancing health outcomes [32,60,63]. This current study found small, significant improvement of KAP scores; however, this impacted the pesticide safety behaviors at home among the participants. For example, before the intervention program, most of the participants did not have a designated room to store the pesticides and farm tools. They also did not keep pesticides in proper areas and did not separate their work clothes from other family’s clothes. After the intervention, they kept pesticides in a closed container and put that in a safe area where children or pets could not reach. They separated work clothes in a basket and put work shoes outside their houses. The knowledge gained from the intervention program, advice from the researcher and monitoring closely by VHVs appeared to influence in-home pesticide safety behaviors.

From this study, the researcher implemented a pesticide training program with community-based participatory research (CBPR). The contents for the training were created based on the issues found from the initial questionnaire. Beyond the receipt of knowledge, the participants were monitored closely, including through a follow-up visit every two weeks and a group meeting every month. This helped the participants to discuss the issues found and encouraged them to share their experiences and opinions of safe pesticide use. After receiving the pesticide safety training program, many of the participants reported and observed proper pesticide practices. Therefore, increasing knowledge and positive attitudes towards pesticide exposure may enhance pesticide safety practices. Of these, it is possible to see improvements in KAP after the intervention.

After adjusting the scores of each perception before the intervention, there was a strong positive relationship between the pre- and post-intervention scores on perceived susceptibility, perceived severity, perceived benefits, and perceived barriers between the groups after the intervention. The results are consistent with a study by Raksanam (2012) using a model to increase agrochemical safety among rice farmers in Thailand [47]. The study found a significant increase in perceived susceptibility and perceived barriers after a six-month intervention. Another study in Iran applied a health belief model and found improvements in the perceived severity of pesticides’ adverse effects, perceived PPE benefits, and cues to action scores among farmers after the intervention [64]. In addition, the study in the U.S. used HBM with La Familia Sana Program (LFSP) and found an increase in scores for pesticide knowledge, the perceived danger of pesticides and the pesticide safety self-efficacy post-intervention [65].

Increasing knowledge can enhance positive attitudes towards perceptions [63]. From this study, the participants gained knowledge from the intervention program. After the intervention, the study found the highest increase in the average scores were the perceived barriers. Most of the participants believed that the use of PPE would minimize exposure. They also believed pesticides are harmful to health and most of them agreed exposure time, duration and frequency, and the amounts of pesticides can result in adverse health outcomes. However, the average scores of perceived benefits slightly increased and some of the participants agreed that the decontamination of pesticides can be improved if taking a bath immediately after returning from their farms. The improvement of knowledge and attitudes after the intervention may lead to an increase in positive beliefs. The construct of the integrated training program is effective and influence positive perceptions among the participants.

Regarding in-home pesticide safety assessment, for the intervention group, the study found a statistically significant improvement of pesticide safety scores (*p*-value < 0.05). The results of the in-home pesticide safety assessment was consistent with Wongwichit [59] and Ajit [48]. These studies revealed an increase in in-home pesticide assessment after receiving training or information related to pesticide safety. In this study, the researcher and VHVs observed the participants keeping pesticides in closed containers and storing them in a safe area where children or pets could not reach. The participants also separated work clothes in a basket and put work shoes outside their houses. The knowledge gained from the intervention program, advice from the researcher and monitoring closely by VHVs appeared to influence in-home pesticide safety behaviors.

As for the control group, the study found an increase in the mean score of in-home safety with borderline statistical significance (*p* = 0.051). However, this study found no significant difference for pesticide safety in this group, other studies also reported comparable results. Lorenz et al. [66] reported an increase in knowledge was associated with an appropriate safety pesticide protocol at homes among pregnant women. The previous studies in Thailand also found a significant association between increased knowledge and proper pesticide practices in maize farmers after the intervention [59] and a statistically significant improvement of the pesticide practices among rice farmers after receiving the agrochemical program [47]. This study found borderline statistically significant in-home safety but not safety about pesticide practices. The participants in this group received one-day training about in-home safety and a manual titled “Clean Home, Good Health, and Happy Life”. It is possible that they gained more knowledge on appropriate in-home safety behaviors. Alternatively, they did not receive pesticide training or information related to pesticide exposure, so knowledge and behaviors would remain the same or show no improvement. However, the results would be stronger if the participants were monitored more closely, such as every two weeks or every month. The results of the checklists could be evidence regarding the effectiveness of the intervention program.

This study has several strengths. Using a community approach helped the researcher to gather insights and useful information from the participants. This also helps the project to continue in the study area. After completing the study, public health workers, VHVs and VLs have agreed to continue the projects. They have planned to distribute pesticide-related information regularly and conduct blood tests every year. As for follow-up visits, the study applied the Line application during the pandemic. This helped the research team maintain social or physical distancing with the participants and the participants were still able to receive essential information related to pesticides improving their knowledge. However, the effectiveness of using the Line application should be determined by further study. In addition, during the pandemic, the participants realized the importance of the study and they could not migrate or move to work in other areas, therefore, all 78 participants completed the study. Furthermore, this was the first study to examine potential reductions in pesticide exposure focused on farm family mothers; therefore, the results of AChE and BChE enzymes can be used as references for other studies.

There were several limitations. Using VHVs to evaluate participants’ homes may lead to observer bias because they know the participants and may positively record the checklists. Moreover, the use of a manual titled “Clean Home, Good Health, and Happy Life” may impact the levels of AChE and BChE in the control group. This is because cleaning a house can indirectly reduce pesticide exposure. The reference of the values of AChE and BChE among female workers aged 18–44 years were limited. This study only found ChE activity among women aged 18–49 years who did not work on the farms and exposure to OPs [67], while another study [55] obtained the training program and assessed the ChE levels after an intervention. This current study did not provide a baseline of ChE activity of the participants before the OPs and CBs exposure, therefore, a baseline before the exposure is needed.

The pandemic impacted the study duration because the researcher was quarantined for about one month, including two weeks in each state, and was faced with local quarantines. Otherwise, this study may have had longer periods for the implementation phase. In addition to COVID-19 regulations, the training sessions could only occur for one day in each area. During the training, the researcher needed to follow the COVID-19 guidelines, making the study more challenging as, for example, wearing a mask creates discomfort and makes it more difficult to communicate among the participants. Moreover, this study may be unlikely to be generalizable to the whole country because of the small scale and the specific area and crops. However, it can be applied to all farmers in Bang Luang, Nan, Thailand as well as other areas that have similar crops and pesticide use.

## 6. Conclusions

Pesticide exposure is one of the main public health issues in Ban Luang, Nan, Thailand. These villagers apply pesticides year-round and are continually exposed to these pesticides, leading to adverse health consequences. The present study suggested that the pesticide training program could increase knowledge, attitudes, practices, and perceptions derived from the health belief model (HBM) among mothers in the intervention groups. Additionally, this program may result in higher levels of BChE in the participants after the intervention. Our methodology and the application of our results can serve as a model for other regions in Thailand and other less developed countries with smaller-scale subsistence and commercial agriculture. 

## Figures and Tables

**Figure 1 ijerph-20-06635-f001:**
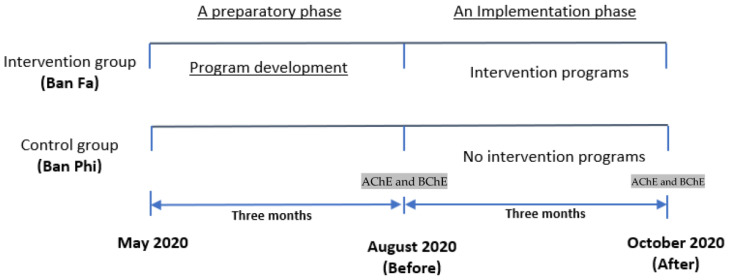
Study design.

**Figure 2 ijerph-20-06635-f002:**
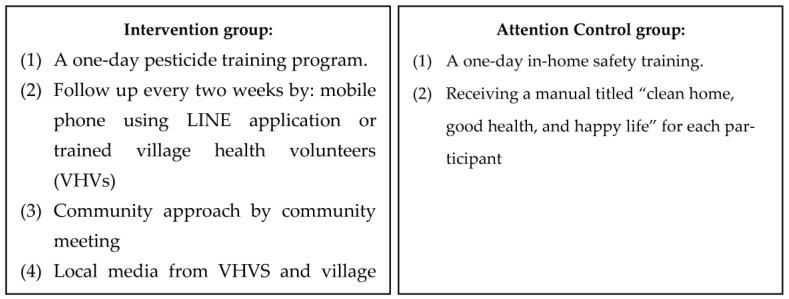
The activities of the intervention and control groups.

**Table 1 ijerph-20-06635-t001:** Demographic and occupational characteristics of the two participant groups before intervention.

Characteristics	Total	Intervention Group (n = 39)	Control Group (n = 39)	*p*-Value
	(N = 78)			
1. Age (years)				0.1 ^a^
- Mean ± SD	36.47 ± 5.58	35.74 ± 4.74	37.21 ± 6.29	
2. Marital Status				0.7 ^b^
- Married	74 (94.29%)	36 (92.3%)	38 (97.4%)	
- Widowed	2 (2.6%)	1 (2.6%)	1 (2.6%)	
- Divorced	2 (2.6%)	2 (5.1%)	0 (0.0%)	
3. Educational attainment				0.4 ^b^
- Primary school	17 (21.8%)	8 (20.5%)	9 (23.1%)	
- Secondary school	21 (26.9%)	8 (20.5%)	13 (33.3%)	
- High school	35 (44.9%)	21 (53.8%)	14 (35.9%)	
- Diploma and higher	5 (6.4%)	2(5.1%)	3 (7.7%)	
4. Income (baht/month)				0.1 ^b^
- 1–6000	59 (75.6%)	33 (84.6%)	26 (66.7%)	
- 6001–12,000	18 (23.1%)	6 (15.4%)	12 (30.8%)	
- More than 12,000	1 (7.9%)	0 (0.0%)	1 (2.6)	
5. Have a room to store farm equipment, fertilizers, and pesticides.				0.2 ^c^
- Yes	21 (26.9%)	13 (33.3%)	8 (20.5%)	
- No	57 (73.1%)	26 (66.7%)	31 (79.5%)	
6. Distance from home to agricultural fields				0.1 ^c^
- ˂50 m	62 (80.8%)	33 (84.6%)	30 (76.9%)	
- ≥50 m	15 (19.2%)	6 (15.4%)	9 (23.1%)	
7. Mixing pesticides				0.5 ^c^
- Yes	25 (32.1%)	11 (28.2%)	14 (35.9%)	
- No	53 (67.9%)	28 (71.8%)	25 (64.1%)	
8. Spraying or carrying hose rubber				0.2 ^c^
- Yes	55 (70.5%)	30 (76.9%)	25 (64.1%)	
- No	23 (29.5%)	9 (23.1%)	14 (35.9%)	
9. Handling pesticides				0.3 ^c^
- Yes	38 (48.7%)	22 (56.4%)	16 (41.0%)	
- No	40 (51.3%)	17 (43.6%)	29 (59.0%)	
10. Planting				0.6 ^c^
- Yes	48 (61.5%)	25 (64.1%)	23 (59.0%)	
- No	30 (38.58%)	14 (35.9%)	15 (41.0%)	
11. Hand harvesting				0.1 ^c^
- Yes	47 (60.3%)	27 (69.2%)	20 (51.3%)	
- No	31 (39.7%)	12 (30.8%)	19 (48.7%	

^a^ independent *t*-test, ^b^ Fisher’s exact test, ^c^ Chi-square.

**Table 2 ijerph-20-06635-t002:** Comparison of mean ± SD of ChE levels between two groups before the intervention.

Variables	Group	Mean ± SD	F	df	*p*-Value
AChE	Intervention	2.7 ± 0.3	0.2	1	0.7
	Control	2.7 ± 0.3			
BChE	Intervention	1.6 ± 0.3	0.4	1	0.6
	Control	1.7 ± 0.3			

ANCOVA test.

**Table 3 ijerph-20-06635-t003:** Comparisons KAP and health belief scores between two groups before intervention.

Variables	Group	Mean ± SD	F	df	*p*-Value
Knowledge	Intervention	36.3 ± 3.4	0.8	76	0.4
	Control	35.7 ± 2.8			
Attitude	Intervention	36.4 ± 3.2	0.2	76	0.9
	Control	36.1 ± 2.6			
Practice	Intervention	38.5 ± 3.1	0.1	76	0.1
	Control	37.9 ± 2.6			
Perceived susceptibility	Intervention	20.97 ± 1.7	0.1	76	0.8
	Control	20.87 ± 1.4			
Perceived severity	Intervention	21.28 ± 1.9	1.8	76	0.2
	Control	21.85 ± 1.7			
Perceived benefits	Intervention	22.28 ± 1.7	0.3	76	0.6
	Control	22.08 ± 1.6			
Perceived barriers	Intervention	22.36 ± 1.9	2.5	76	0.1
	Control	21.74 ± 1.4			

ANCOVA test. Note: Scores of knowledge, attitude, and practice range from 0–50, 16–48 and 0–54, respectively. Scores for each perceived category range from 6–30.

**Table 4 ijerph-20-06635-t004:** Intervention effects and effectiveness of the program on ChE activity between groups after intervention.

Variables	Group	Time	Mean ± SD	F	df	*p*-Value	Partial Eta Squared
AChE	Intervention	Before	2.7 ± 0.3	1.2	1	0.3	0.02
		After	2.8 ± 0.2				
	Control	Before	2.7 ± 0.3				
		After	2.7 ± 0.2				
BChE	Intervention	Before	1.6 ± 0.3	11.6	1	<0.0	0.20
		After	1.8 ± 0.2				
	Control	Before	1.7 ± 0.3				
		After	1.7 ± 0.2				

ANCOVA test.

**Table 5 ijerph-20-06635-t005:** Intervention effects and effectiveness of the program on the KAP and health beliefs derived from HBM.

Variables	Group	Time	Mean ± SD	F	df	*p*-Value	Partial Eta Squared
Knowledge	Intervention	Before	36.3 ± 3.4	7.8	75	<0.0	0.10
		After	38.7 ± 4.3				
	Control	Before	35.7 ± 2.8				
		After	36.1 ± 3.9				
Attitude	Intervention	Before	36.4 ± 3.2l	33.7	75	<0.0	0.31
		After	38.0 ± 3.1				
	Control	Before	36.1 ± 2.6				
		After	35.9 ± 2.5				
Practice	Intervention	Before	38.5 ± 3.1	28.2	75	<0.0	0.27
		After	39.3 ± 2.8				
	Control	Before	37.9 ± 2.6				
		After	37.5 ± 2.4				
Perceived susceptibility	Intervention	Before	20.9 ± 1.7	37.6	75	<0.0	0.33
		After	23.6 ± 1.6				
	Control	Before	20.8 ± 1.4				
		After	21.5 ± 1.8				
Perceived severity	Intervention	Before	21.3 ± 1.9	34.5	75	<0.0	0.38
		After	23.8 ± 1.4				
	Control	Before	21.9 ± 1.7				
		After	22.0 ± 1.6				
Perceived benefits	Intervention	Before	22.3 ± 1.7	38.8	75	<0.0	0.34
		After	23.5 ± 1.3				
	Control	Before	22.1 ± 1.6				
		After	21.6 ± 1.7				
Perceived barriers	Intervention	Before	22.4 ± 1.9	25.6	75	<0.0	0.26
		After	23.7 ± 1.5				
	Control	Before	21.7 ± 1.4				
		After	21.8 ± 1.4				

ANCOVA test. Note: scores of knowledge, attitude, and practice range from 0–50, 16–48 and 0–54, respectively. Scores of each perceived range from 6–30.

**Table 6 ijerph-20-06635-t006:** Pairwise comparison assessments of pesticide safety and in-home safety within groups.

Program	Group	Time	Mean ± SD	95% CI Difference		*t*	df	*p*-Value
				Lower	Upper			
Pesticide Safety	Intervention	Before	8.0 ± 1.3	−0.9	−0.1	−2.5	38	0.0 ^e^
		After	8.5 ± 1.0					
	Control	Before	8.1 ± 1.0	−0.5	0.2	−0.9	38	0.4
		After	8.3 ± 0.9					
Safety In-home	Intervention	Before	7.8 ± 1.4	−0.5	0.1	−1.2	38	0.3
		After	8.0 ± 1.0					
	Control	Before	7.3 ± 1.1	−0.4	0.2	−0.9	38	0.051 **
		After	7.7 ± 1.1					

^e^ Paired *t*-test, ** *p* = 0.051.

## Data Availability

Data are available on request.

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
