# Peer review of "An Evaluation of a Pesticide Training Program to Reduce Pesticide Exposure and Enhance Safety among Female Farmworkers in Nan, Thailand"

_ijerph, 2023, doi:10.3390/ijerph20176635_

Round 1

Reviewer 1 Report

This manuscript examined the effect of a pesticide training program on the blood pesticide exposure level and safety attitudes among female farmworkers in an agricultural area in Thailand. The work could be improved by more rigorous presentations of the results, as well as some explanations and clarifications:

-Introduction:

The introduction could have been improved by adding some background information to show why it is important to focus on female farmworkers with young children in this study and what kinds of health outcomes could have been caused by pesticide exposures to females and children compared with male farmworkers.

-Study design:

One sub-district (Ban Fa) was selected due to high exposure level and exposure related health outcomes, while the other sub-district (Ban Phi) was previously selected as control area. It is unknown how the control group was selected.  

-Quasi-experiment:

The authors mentioned that this was a quasi-experiment, and a random sampling system was used to select participants among eligible villagers who were willing to participants. However, in a quasi-experiment, researchers often do not have control over treatment assignment but use pre-existing groups that receive different interventions after the fact. The study design is confusing.

-Follow-up visit:

The follow-up visits were conducted in October 2020. It was unknown whether blood test results and questionnaire/health beliefs responses only reflected short-term effect of the pesticide training program. It would be interesting to know long-term effects of the program.

Since the pesticide program was shown to be useful to reduce pesticide exposure level and improve health attitudes among the study participants, it would be beneficial to also provide the same pesticide training to the control group as well as individuals who were eligible but not selected to participate in the study after the intervention. Usually in a step-wedged study design eventually everybody including the control group gets the same beneficial intervention. It would be great if the authors could expand some discussions on long term procedures.

It was not mentioned whether individuals in the two intervention arms were blinded, i.e. whether they were able to tell which group they were assigned to and which program they were provided compared with the other group, and how this could have impacted the study results.

-ANCOVA:

In Table 2 and 3, the intervention and control groups were compared using the ANCOVA test. An ANCOVA test is often used when there are more than two groups and covariates are included in the test. It is not clear why an ANCOVA test was conducted here when there were only two groups and which covariates were adjusted for.

 Similarly, in Table 4 there were only two groups to compare with, it is confusing why an ANCOVA test was used. Instead a regression model to model post-intervention AChE and BChE level on group (intervention/control) and pre-intervention AChE and BChE level would be convenient.

-When the observed difference between two groups was mainly due to random, a non-significant P-value was also provided. For example, the author mentioned “The participants (36.3+-3.4) in the intervention group had higher scores in the knowledges compared to the control group (35.7+-2.8)”. Instead of interpreting as “higher/lower” score, it should just be interpreted as “similar” or “no difference”.

Some grammar needs to be improved. 

Reviewer 2 Report

Please see the attached report for full comments and suggested revisions.
